# Expertise based skills management system to support resource allocation

**Nazia Bibi<sup></sup>, Zeeshan Anwar⦿<sup></sup>*, Tauseef Rana<sup></sup>**

Department of Computer Software Engineering, National University of Sciences and Technology, Islamabad, Pakistan

⦿ These authors contributed equally to this work.
* zeeshan.phdcse@students.mcs.edu.pk

## Abstract

Skills Management is an essential concept of human resource management in which a skill inventory may be created for each employee and managers can assign tasks to workers based on worker's abilities. This concept is not fully practiced for two reasons: i) employee's skills are not effectively evaluated and documented, ii) tool support is deficient to manage this complex task. Ineffective skill management of an organization fizzle tasks assigned to the incompetent employees and this may lead to project failure. To fill up this gap, a survey is conducted across various software organizations to find out the best practices for the skill management and to gather requirements for skills management framework. Based on survey findings, a mathematical framework is proposed that calculates the soft and hard skills of employees automatically based on time and achievements as skill increases or decreases over time. In this framework, the Skills Calculation Engine (SCE) is developed for the managers to enhance the capacity of appropriate decisions making in assigning tasks to the rightly skilled workers. This framework is also useful for organizations as it can increase profitability as tasks are assigned to the most appropriate employees. The SCE is implemented as a Windows-based application to calculate skills, store skills in skills inventory, and assign tasks based on an employee's skills. The skills management tool is evaluated in a facilitated workshop; furthermore, a feature-wise comparison of the tool is also made with existing tools.

**Data Availability Statement:** Data can and code is available online from author github repository. We have mentioned the github repository in paper. The address of repository is as follow: https://github.com/zeeshan0333/SkillsManagement.

## 1 Introduction

A Company must be acquainted with the skills of the employees so that the right person can be selected for the right job. Researchers [1] identified that software development/maintenance projects either in-house or outsourced will only be redsuccessful if appropriate human resources are engaged in it. If the organizations fail to assign suitable human resources in project activities then these organizations will face constant failure [2]. Hence, for the success of a project, appropriate tasking of employees is very crucial [3]. In many software development organizations, the required skill assessment system is not being implemented due to lack of awareness of its strengths [4].

**Funding:** No funding was taken for this research. The funders had no role in study design, data collection and analysis, decision to publish, or preparation of the manuscript.

**Competing interests:** The authors have declared that no competing interests exist.

Software projects are different from other discipline's projects because the success of software projects depends on how effectively and efficiently human skills are utilized [4]. Unfortunately, there is no broad approach and tool support for monitoring and evaluating human skills [5–7]. Researchers agreed that there is a severe need for calculating human skills [6], but due to the un-availability of skill assessment systems, organizations are losing time and money. Due to competitive pressures in the industry, it is obligatory to adopt the Decision Support Systems (DSS) for skill assessment. By adapting computerized DSS, organizations will be able to utilize time efficiently in decision making and to perform tasks in a better way [8].

Employee's skills are overlooked in the software development industry because determining relationships between employee's skills and software development is a complex task [9]. As a result, assigning an employee to an appropriate task is hard to put into practice in big organizations [10]. Due to large-scale human resources, it is challenging to update the skills of all individuals. Global firms are facing more difficulty in managing employee's skills efficiently [11]. To tackle such problems, it is very decisive to maintain detailed information about the employee's qualifications, experiences, and expertise. It is nearly impossible for the managers to retain a comprehensive profile of all employees without an effective software support [12]. Similarly, managers are unable to memorize the employee's skills and knowledge for assigning specific tasks to skilled employees.

The need for computerized skills evaluation is increasing day by day because using a tool to manage skills improves decisions precision [13, 14]. Researchers explored this fact saying that taking the right decisions is very important for the success of software projects/ industry [15]. Researchers are now focusing on multiple skills of individuals during project scheduling [16]. A pool of multi-skilled employees is available, but allocating a particular employee to a specific task requires consideration of enormous parameters like Product Development Life Cycle (PDLC) time, skill efficiency gain and cost [17].

Above in view, it is the need of every organization to assign tasks to employees, according to the skills they possess. But it is very challenging to find which project task is suitable for whom. As skill calculation is a difficult task, so most of the time managers fail to assign tasks to the employees according to their skills [13, 14]. In this paper, we proposed a skills calculation framework and it calculates the skills of employees before assigning tasks. In this framework, the skills are calculated by considering various factors which include: responsibilities assigned to an individual, the task performed by an individual, education, training, and emotional intelligence (ability to control emotions and achieve goals). The proposed framework automatically increase or decrease the rating of skills based on employee's performance, experience, and qualification. It provides benefits to the organizations, management, and employees. Managers can calculate the skills of the employees; employees get tasks according to their skills and will be in a better position to perform well. The major contributions of this research are as follows:

- A survey is conducted across various organizations to understand the existing practices for skills management and to gather requirements to develop the skills calculation framework.

- An algorithm and a mathematical framework are proposed that calculates the skills of employees based on their profile. The algorithm will increase or decrease the skill over time.

- A prototype software tool is developed to automate redskill calculation and skill inventory management. The software is evaluated by various experts. We uploaded our tool online for future research and improvement.

- Comparison of the prototype software tool with existing literature and the existing software applications [18–20] is also made.

In the rest of the paper, Related work is given in Section 2. In Section 3 our research methodology is given. An overview of the proposed framework to calculate skills is provided in Section 4. Section 5 covers the system implementation details. Evaluation and Comparison of our tool are presented in Section 6. Finally, the conclusion and future work are presented in Section 7.

## 2 Related work

For knowledge intensive organizations, human resource management is one of the key factors for success. To find the right person for the right job is of prime importance and is one of the major issues for organizations. Managers assign projects to those employees whose skills are relevant to the project. Human skills are important because most of the studies have proved the fact that employee's skills should be relevant and contributes towards the success of the project [10, 21, 22].

Kerzner [23] discussed that evaluating and balancing employee's skills require a lot of effort. The decision to utilize either task skills or people skills is the major pitfall. Employee's assignment to tasks should be proportionate with their skills, otherwise, this may lead the project towards failure. Badaracco et al. [24] proposed a procedure for the item using a multi-criteria decision model for competence training. Computerized Adaptive Tests (CAT) adaptation accuracy to the student's proficiency level is improved by the proposed approach. This approach is not suitable when it comes to the applicability to different sectors. In the meantime, tests are adapted to the students' competencies, but the competency analysis to map and fill competency gaps is not supported. Therefore, a generalized approach is needed for the evaluation of skills required in projects because numerous types of human skills are required in projects and these skills need to be updated constantly.

For the better performance of the firms, a proper assortment of expertise is very decisive. Researchers focused on two main approaches for the selection of experts, i.e. relative methods and absolute methods [19, 25–27]. It is evident from literature [26] that certain criteria are used for defining levels of expertise. These criteria can be written criteria or professional achievement [28]. The European Commission, 2008, in its work explores the literature which aims to enrich the definition of competence and to cover all potential criteria for competence management. However, there is no practical implementation of the concept. As a consequence, their work offers a very good theoretical basis and foundation for our research area. The integration of scientific methods/algorithms in this regard will add significant additional benefits to the results of this study [29].

Wang et al. [30] proposed multi-objective algorithm which addresses the scheduling problem of multi-skill resource. The idea is to minimize the cost during the scheduling process. The algorithm takes the advantage of knowledge to perform sequential search and to reassign and readjust the resources to respective tasks. Although this approach is useful, but the emotional intelligence factor is ignored during multi-skilled resource scheduling.

Huan et al. [31] utilized a multi-purpose evolutionary technique to optimize the expansion of competency sets by multiple criteria. This works lacks in term of flexibility as the adaptation of their proposed approach to a different range of areas is not adequate.

Ahmed et al. [16] also identified the fact that many organizations face serious problems because project managers are unable to understand the project's requirements and the skills required to accomplish the project. Their work provides a theoretical basis of the skill required in different phases of software development [16, 32]. The theoretical basis is not adequate and requires skill management system. To solve such problems efficient utilization of the resources

and a skill management is crucial because every individual is not fit for every task as they have different abilities and traits.

Amiri et al. [33] proposed approach is based on a hybrid model which uses multi-criteria decision making to assess the company's expertise. The approach also provides practical results of implementation. However, it lacks generality and flexibility in various fields and failed to give a basis for the competences model. The scope of application is also different from this work as it applies to the capabilities/competencies of companies rather than individuals.

Olsen [34] explored that to accomplish certain tasks, skill management tools and techniques should be efficiently managed and implemented. Lee et al. [35] and Lin et al. [36], work includes the core competencies of technical experts for integrated circuit design and research and development. Both areas require necessary expertise in high-performance. Different approaches to competencies management were investigated in both papers, but the practical implementation and application of algorithms are missing.

A job description is a common approach for human skill management. Job evaluation is possible by using skill level and other factors of employees [37]. In most organizations, employees themselves rate and maintain their information. If they rate themselves too high, they can face some difficulties because if they fail to meet targets or fail to answer questions about their high ratings, their reputation will be suffered in front of their colleagues. A computer-based test method developed in the project "Measuring Experimental Competences in the Large Scale Assessments" [38] allows accurate and reliable measurement of experimental capabilities comprehensively. "Technology-based Assessment of Skills and Competencies in VET (ASCOT)" [39] as an initiative consists of 21 projects covering six key areas and requires close collaboration between research institutes, VET professionals, and facilities. Most of the projects in this initiative are theoretical studies that do not apply computer methods and are limited to academic skills without application. At the same time, these projects focus only on the evaluation phase, rather than sorting and matching job applicants and the selection of qualified employees.

Furthermore, researchers analyzed [40] that it is the responsibility of the project management team to analyze the project output, accomplishment, and fulfillment of the goals. For obtaining these goals, some internal performance measures must be implemented [41]. But assessing the performance of the projects is very crucial [42]. Jiang and Klein also explored that performance can be measured through output level, time management and the actual cost of the project [43]. Although accomplishing project success is possible only, if the workers/employees linked with the project are skillful and specialists in particular tasks. A skill evaluation is needed before assigning any task to an employee; otherwise, this would impact in project's productivity.

The quality of software development depends on the skills of developers. Therefore, it is really important for PMs to first determine the proper skill set and then assign activities for said purpose, Chandsarkar et al. [18] built a small application for PMs to efficiently organize and retrieve resources. This application retrieves employees with the best skillset and if an exact match is not found then it provides the closest match.

The project manager needs an automated skill evaluation system. But unfortunately, organizations are not utilizing skill management systems properly because the skills and qualifications of employees cannot be determined by a single tool. Hence, incorporating these two factors in a tool increases the production and profitability of an organization [21].

In order to keep a detailed profile and to track the skills and qualifications of employees, it is very important to use the skills management system. Because it helps in finding suitable and skilled employees with appropriate qualities and abilities and then it is easy to position an individual for different tasks. The human resource department also uses this system to match the

skills of employees for certain tasks and then assign tasks accordingly. Moreover, it saves time and costs of the overall project [16, 32].

Project managers believe in continuous improvements in technical abilities and for this training and development programs need to be promoted to bring improvement at the organizational level [16, 44]. The training should be arranged by keeping in mind that particular skills need to be enhanced. Training, developments, and technical improvements will bring positive change in the organization in terms of the improved capabilities of employees, task accomplishment and flexible workforce environment [44, 45].

The approach proposed supports the requirements of Competence Management System (CMS) as found in the general definitions of competence and CM (Competence Management) in this section. These requirements in particular are considered in establishing the reference mathematical model. Overall, an in-depth review of the literature makes it clear that CMS demands a scientific and workable mathematical approach to improve job performance in an organization. It can be observed in the literature review, previous efforts in this area have come primarily from an HRM and psychological perspective, relying on paper-based and traditional practices, and deficient integration of computer science methods with CMs.

Furthermore, after the literature review it has been concluded that any EBSMs at least includes: (1) required competencies identification (exploration of competencies), (2) acquired competencies/skill evaluation (capability assessment), (3) analysis of competencies/skills (required and acquired skills matching) and (4) provide recommendations for further capacity development planning and improvement of competencies/skills gaps (capacity development). Most of the research projects reviewed used traditional assessment methods for competencies. As mentioned earlier, these methods have difficulties in adopting such an approach to new fields (careers).

In most of the literature, it is discussed that one of the main challenges is the lack of a generalized and efficient competence matching method. Considering the European refugee crisis and the main problem of skill mismatch, this work effectively encourages e-recruitment in the form of skill discovery and matching solutions. The proposed EBSMs can be adopted easily for software organizations because it provides a mathematical basis for evaluating competences.

## 3 Research methodology

Our research methodology as shown in Fig 1 is divided into four major steps i.e. survey, development of the framework, implementation of the framework, and finally validation of framework. Details of each step are given below:

As the motivation for the proposed framework in the paper, we have gathered data through questioners [46] from the various software organizations. In this survey, the forms are filled by Team Leads, Project Managers, HR Managers, Program Managers, Testers, CEOs, Developers, CTOs, Designers, and System Analysts. We anonymized the identity and organizations of survey respondents as the identity of respondents is not used and required in our research. The sampling method we used is convenient sampling because we want to gather requirements from all employees. The survey is conducted in a short time using this technique. The questionnaire is designed after exhaustive brainstorming and literature review [5, 7, 9, 30, 37] to collect information about the expertise of an individual working in the software development industry. This questionnaire is categorized into five sections as given below and contains 28 direct and indirect questions.

- Concept verification—The first section of the questionnaire verifies the concepts that are base of this paper. We have asked different questions to verify that either we are moving in

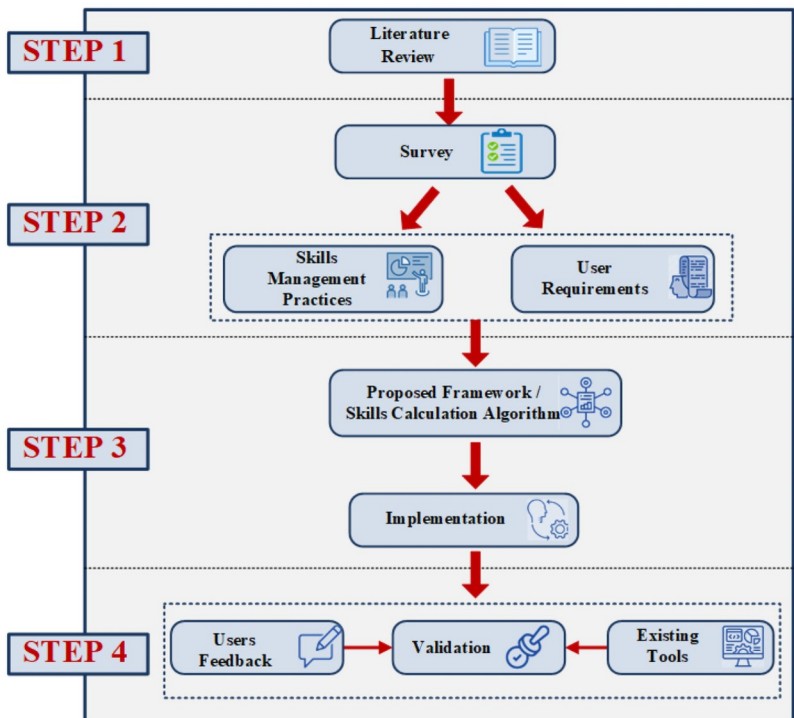

**Fig 1. Research methodology.**

the right direction or not. These questions are related to the verification of multiple skills concepts, skills rating mechanism, and practices.

- Skill Rating—This section presents various skill rating methods to get feedback from the survey respondents about the best method for the software development industry that should be opted to rate skills.

- Resource Allocation Practices—This section covers questions related to human resource allocation practices. Resources are allocated based on the job description, expertise, or success rate.

- Relation between Resource Allocation and Skill Management—This section finds and explores the effectiveness of the skill management concept for resource allocation. Many questions have been asked from the survey respondents about what are their expectations from a system/tool while allocating resources to projects. Respondents have been asked to give feedback on the tool support in the software development industry to evaluate/rate employees based on expertise. The tool should have functionality to facilitate the resource allocation process.

- Expertise Based Skill Management (EBSM) Requirements—This section of the questionnaire collects user's requirements and their expectations about the EBSM model. Feedback about the integration of EBSM with project management software is also gathered.

After analyzing the results of the survey, we developed the mathematical framework to calculate the skills. The framework is implemented as a windows application in C# language. SQL Server is used to store the data. The tool is evaluated by using three methods i.e. i) Questionnaire; ii) Facilitated Workshop and iii) Comparison of our prototype tool with existing tools.

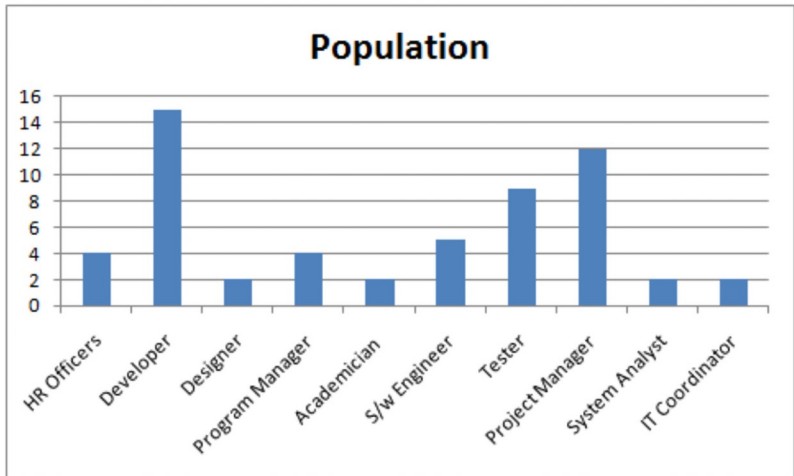

**Fig 2. Population of study.**

## 4 The proposed framework

A framework to improve the skills management as shown in Fig 3 is proposed after analyzing the results [46] of the survey. The proposed framework covers the limitations of Feng et al. [7] which states that employee's skills are not consistent with the management practices of software development organizations. The population of the study as given in Fig 2 is employees of software companies. The majority of the population is developers and project managers. We distributed 200 questionnaires out of which we get 57 responses, the response rate is 28.5%. The responses help us to figure out the important requirements that are considered for developing the framework. Highlights of the survey results are given as under:

*Concept Verification*: 82.45% marked agree or strongly agree by the statement that *Employees have many skills other than their job description*. The idea that skills should be rated based on proficiency is agreed upon by 89.47% people. 68.42% participants admit to utilizing the extra skills of their subordinates. 42.1% people believe that emotional intelligence can serve as an adequate measure of the proficiency and skill of employees.

*Existing Practice for Project Resource Allocation (PRA)*: 48% respondents do skill management based on the job description. They use the experience of an employee as a measure of his/ her skill level. 50% have the desire to gauge the skills using measurements of performance. Currently, PRA practice uses a Job Description and Success Rate based approach.

*Resource Allocation Problem*: 43% struggle with finding suitable employees to give project work. 30% report that they have a shortage of skills record.

*Unrelated Jobs Assignment*: 46% people recruited are not suitable human resources (employees) and then assigning jobs to the recruited resources which are not relevant or beyond their expertise.

*Skill Management (SM) and Resource Allocation (RA)*: 77.19% do not use any kind of skill management software, while 40.35% of people feel that they do not require any such tool. 84.21% people have the belief that project resource utilization will be positively impacted by skill management. 91.22% people think that an organization's overall performance can be increased with aid from skill management. 96.49% of people think that project management

will be helped by adequate skill management. The EBSM approach is chosen by 82.45% of people while 89.47% of people wish to integrate the skill management in the project management software.

*Integration with Project Management Software*: 36% of people wish to view the record of the skills in the resource sheet, while 40% expect that there should be a separate view for skills and 24% think that skills should be visible in the resource calendar.

*EBSM Model Requirements*: The requirements for most people include; module to rate skills, module to enter educational data, the record of success/failure of current and past projects, training record of employees, job description record of employees, and seamless integration with project management software.

Fig 3 shows the conceptual model of the framework of the product proposed in this paper. It addresses the requirements for calculating the expertise and skills of an employee, supports the project manager in decision making for assigning the job to the most suitable and qualified skilled personnel. The framework is implemented as a tool to facilitate the decision-making process by calculating the skills of an employee working within the organization. Requirements for the framework are captured from each section of the survey. From the concept verification section, it is validated that employees possess multiple skills. The skill calculation is based on training, responsibilities / experience, performance, educational record, and emotional intelligence. Therefore, we include these features in the framework to calculate the skills. The framework should rate the skills on a scale of 1 to 10 based on the feedback of the participants. Currently, managers use expert judgment to measure the skills; therefore, we designed a Decision Support System (DSS) to facilitate managers to rate employee's skills. The DSS rate each skill of employees; whereas, the performance management system rates the employee's overall performance.

From the existing practice for RA and resource allocation problem section of the questionnaire, we get more information about resource allocation and the root cause of the problem in assigning resources. The resources are assigned based on their success rate. The proposed framework helps project managers to find suitable employees for the tasks because the project manager will have the information of all employee's skills along with skill ratings.

The next three sections of the questionnaire, provide information about the usefulness of the skills management system and user requirements/features that should be included in the framework. Most of the participants want to use skills management systems in their organizations and the various features to be included in the framework which are given in Table 1.

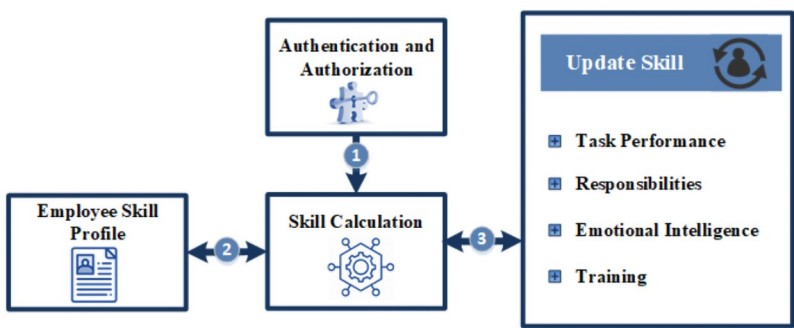

**Fig 3. Proposed framework.**

**Table 1. Features of framework.**

| Sr. No. | Features |
|---|---|
| 1 | Job description record |
| 2 | Employee experience/profiling module |
| 3 | Training record |
| 4 | Success rate of employee |
| 5 | Skill rating facility |
| 6 | Emotional Intelligence record |
| 7 | Expertise based rating of skills |
| 8 | Integration with project management software |

## 4.1 Skills calculation algorithm

We devise a skill calculation algorithm after extensive literature review [5, 7, 9, 30, 37] and getting feedback from practitioners in the form of a questionnaire. Skills calculation consists of six steps. As proposed in the existing work [7], we also included problem-solving and analytical ability in skills calculating algorithm. All steps of the skill calculation the algorithm are given in Fig 4. The description of notations used in the algorithm is given in Table 2.

*Step 1*: In the first step; we created the skill matrix which consists of various skill items that are required in an organization. Levels are defined from 1 to 10 for each skill item; where 1 means lowest and 10 means highest. The number of skill items can be represented by a matrix as shown in Eq (1).

$$S_{\text{Initial}} = \{S, M_i, B, AB, PS\} \tag{1}$$

Where $S_{\text{initial}}$ is the initial skill matrix, S = Specialization in a major subject, $M_i$ = A set of minor subjects (i.e. M1, M2, M3,...), B = Behavior rating, AB = Analytical ability and thinking, PS = Problem solving. Each feature is shown on the right-hand side of Eq (1) number to represent the skill level.

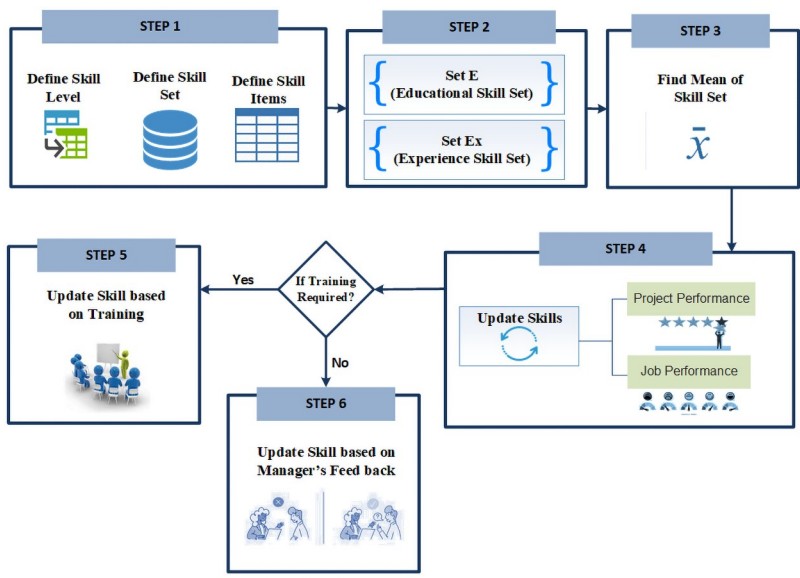

**Fig 4. Skills calculation algorithm.**

**Table 2. Algorithm symbols representation.**

| Notation | Description |
|---|---|
| $E_F$ | Employee's experience in a subject |
| SL | Skill Level |
| $S_{Initial}$ | Initial Skill Items/matrix |
| BR | Behavioral Rating |
| $BR_S$ | Satisfied behavior |
| $BR_G$ | Good behavior |
| $BR_{Exc}$ | Excellent behavior |
| $(E_F)_{Beginner}$ | Employee is a Beginner |
| $(E_F)_{Intermediate}$ | Employee is not Beginner or Expert |
| $(E_F)_{Expert}$ | Employee is an Expert |
| $SL_{EX}$ | Skill rating based on experience |
| $SL_E$ | Skill rating based on Education |
| PM | Project Manager |
| $T_{Set}$ | Training Set |
| $EI_S$ | Satisfied emotional Intelligence Level |
| $EI_G$ | Good emotional Intelligence Level |
| $EI_{Exc}$ | Excellent emotional Intelligence Level |
| PS | Problem solving Skill |
| AB | Analytical ability |

$S_{initial}$ is populated with the information given by the Employee. This initial matrix will be updated later based on the employee's education and behavioral record. Analytical ability and problem-solving skills are important to consider before job assignments along with technical expertise. An employee with these skills can deal with the dynamic and challenging environment [47].

*Step 2*: In step 2; we created the two subsets SetE (Educational Skill Set) and SetEx (Experience Skill Set). SetE is calculated based on the educational record of the employee. The highest rating is given to major subjects. If the employee has previous experience, his $Set_{Ex}$ (as shown in Eq (5)) is created and skills are rated based on working years for each subject as shown in Eq (2). Only the main field(s) of employees will be rated in this step.

$$SL_{Ex} = \begin{cases} (E_F)_{Beginner} = 1 & \text{if } 1 \leq x < 5; or \\ (E_F)_{Intermediate} = 5 & \text{if } 5 \leq x \leq 9; or \\ (E_F)_{Expert} = 10 & \text{if } x \geq 10 \end{cases} \tag{2}$$

$$BR = \begin{cases} BR_s = 1 & \text{if } 1 \leq x < 5; or \\ BR_G = 5 & \text{if } 5 \leq x \leq 9; or \\ BR_{Exc} = 10 & \text{if } x \geq 10 \end{cases} \tag{3}$$

Eq (3) is used to calculate the Behavioral Rating of an employee on a scale of 1 to 10.

$$SL_{Ex} = \frac{SL_{Ex} + BR}{2} \tag{4}$$

Eq (4) is used to take mean value of $SL_{Ex}$ and BR. Here we added BR because we consider the attitude of an employee towards work. So, the employee's experience level is also updated based on the behavior.

$$Set_{Ex} = \{SL_{Ex1}, SL_{Ex2}, SL_{Ex3}, \ldots \ldots, SL_{Exn}\} \qquad (5)$$

The $Set_{Ex}$ matrix is formed using Eqs (2) and (3), which measures skill level of an employee in each subject/field based on experience and behavior. If an employee has experience of 1 to 5 years, the employee is assigned the value 1 which means he is at beginner level. If an employee has an experience of 5 or more than 5 years but less than or equal to 9 years then value 5 is assigned, which means he is at the intermediate level. Similarly, if an employee has experience of 10 years or more then PM assigns the value 10, which means that the employee is an expert. Eq (5) shows the skills rating of an employee in each subject/field based on his experience, Where $SL_{Ex1}$ is an employee's experience in subject/field 1 and $SL_{Exn}$ is employee's experience in subject/field n.

It is important to take the mean value as each item is rated on a scale of 1 to 10. So, Eq (4) is used to take the mean/average value.

$$SL_E = \begin{cases} S = v & Z1; or \\ M_i = v_i & Z2 \end{cases} \qquad (6)$$

where Z1 = Project Manager (PM) assigned value (v) to the specialized subject on a scale of 1 to 10 and Z2 = PM assigned value v to minor subjects on a scale of 1 to 10 for a given employee.

$$Set_E = \{S, M_i\} \qquad (7)$$

The $Set_E$ matrix is formed using Eq (6), which measures skill level based on an employee's education. A manager rates major subject (S) and minor subject ($M_i$) on a scale of 1 to 10 but a higher value is assigned to specialized subjects as compared with minor subjects. Eq (7) shows the skill rating of employees based on educational records. The educational set covers the skills that an employee learns in university [6].

*Step 3*: In step 3, SetE and SetEx is merged and the database is populated with the mean value of SetE and SetEx.

$$Skill_{Rating} = \frac{Set_{Ev} + Set_{Exv}}{2} \qquad (8)$$

*Step 4*: The input to step 4 is an initial skill matrix ($S_{Initial}$) of an employee that is created at the configuration of software, refer to step 1 of Fig 4. In step 4, the skills of the employee will be updated (rated higher or lower) by the manager based on the performance of the employee using Eq (9). Here the performance of an employee related to project activities, i.e. soft/hard skills, and performance at the workplace, i.e. attitude, punctuality, soft skills, etc. will be rated. Eq (9) will be automatically updated by software.

$$Skill_{Set} = update(Skill_{Set}) \qquad (9)$$

*Step* 5: In this step; the skills of an employee will be updated based on training and a rating will be given based on training evaluation.

$$T_{\text{Set}} = \begin{cases} E_{ij} = 1 & \text{if } i^{th} \text{ employee has } j^{th} \text{ training,} \\ \\ E_{ij} = 0 & \text{Otherwise} \end{cases} \tag{10}$$

$$AB = \begin{cases} AB_s = 1 & \text{if } 1 \leq AB_s < 5; or \\ AB_G = 5 & \text{if } 5 \leq CT_G \leq 9; or \\ AB_{\text{Exc}} = 10 & \text{if } AB_{\text{Exc}} \geq 10 \end{cases} \tag{11}$$

$$PS = \begin{cases} PS_s = 1 & \text{if } 1 \leq PS_s < 5; or \\ PS_G = 5 & \text{if } 5 \leq PS_G \leq 9; or \\ PS_{\text{Exc}} = 10 & \text{if } PS_{\text{Exc}} \geq 10 \end{cases} \tag{12}$$

Eqs (11) and (12) are used to rate analytical ability and problem solving skill respectively.

*Step* 6: In this step; the manager will update the skills set based on the emotional intelligence and overall performance of an employee in the organization.

$$EI = \begin{cases} EI_s = 0.5 & \text{if } 1 \leq x < 5; or \\ EI_s = 0.8 & \text{if } 5 \leq x \leq 9 or \\ EI_{\text{Exc}} = 1 & \text{if } x \geq 10 \end{cases} \tag{13}$$

Eq (13) is used to measure levels of emotional intelligence of an employee.

*Step* 7: The output of steps 5 and 6 generate the final skill set of an employee shown in Eq (14). This skill set is continuously updated as steps 4 to 6 will be repeated based on the Standard Operating Procedures (SoPs) of the organization.

$$Skill_{\text{Set}} = update(Skill_{\text{Set}} * EIRating) \tag{14}$$

## 5 System implementation

As a proof of concept, we have built a tool for skill management, which implements the proposed algorithm as given in Fig 4. Our prototype tool solves the problem of skill calculation. More specifically, the personnel database and skills inventory of the employee can be managed. In addition to manipulating personal data, authentication and authorization mechanism are the features of our system. After successful log-in the Admin can assess skills; update the employee's skills based on his performance of assigned tasks, responsibilities, training, and emotional intelligence rating. The manager can also add, update, delete and view the projects and associated tasks. He can assign tasks to employees based on their skill set ratings as he can view and generate a skill matrix of employees. Whereas, an employee can only add, update his personal information, education, experiences, and training. Admin can add skill set and training information required by the company.

As creating a project plan is a very critical activity for a project manager. Tempting to make a project schedule and complete tasks as soon as possible is the manager's topmost priority. For this purpose, a manager needs some automated tools. MS project is a quite popular software for managing resources, develop project plans, manage budgets, create schedules and

track progress. Although MS project is a good tool for scheduling but leveling of resources is an issue while working in MS Project and secondly, it does not support skill calculation. One can manually calculate employee skills and then assign resources to project activities [48]. This ends up with poor management and results in inappropriate human resource allocation especially in large organizations.

With our proposed approach (the framework and the supporting tool), a project manager evaluates employee competencies because project activities/tasks required different skills of an employee. This tool provides skill rating based on employee experience and job performance, which helps PM during the assignment of employees to the best-fitted task by considering employee EI (Emotional Intelligence) factor. Emotional intelligence is one measure of human resource competencies. According to the Marie-Louise Barry survey [4, 49], 89.8 percent of respondents agree that EI is necessary for PMs.

## 5.1 Working environment

This system runs on a Windows platform and is developed on a computer running Windows 8 operating system. The front end of the system is developed in Visual Studio and the database is stored in SQL Server. We have uploaded the code of the project and the database on our GitHub project page [46]. Anyone can extend the project for research purposes.

## 5.2 Major modules

The system consists of eight major modules and different forms for user interactions. It is not possible here to list the features of each module. Therefore, we have uploaded the complete requirement specifications and screen shorts on our GitHub project page [46]. Some of the screen shorts of form are given in Figs 5–8. The modules can be categorized based on the functionality and user's role. Here, we give a brief description of each module based on the functionality of the module.

**5.2.1 Authentication and authorization.** This module authenticates the user and categorizes the user into three roles, i.e. manager, employee, and administrator. After successful login, users can use the system, and a connection with the database is established. Interfaces are also divided into three classes, these classes are based on the roles of the user.

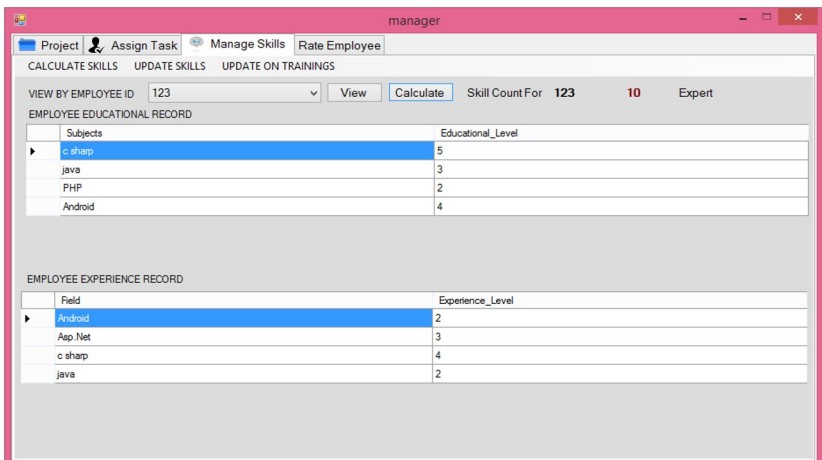

**Fig 5. Employee skill calculation.**

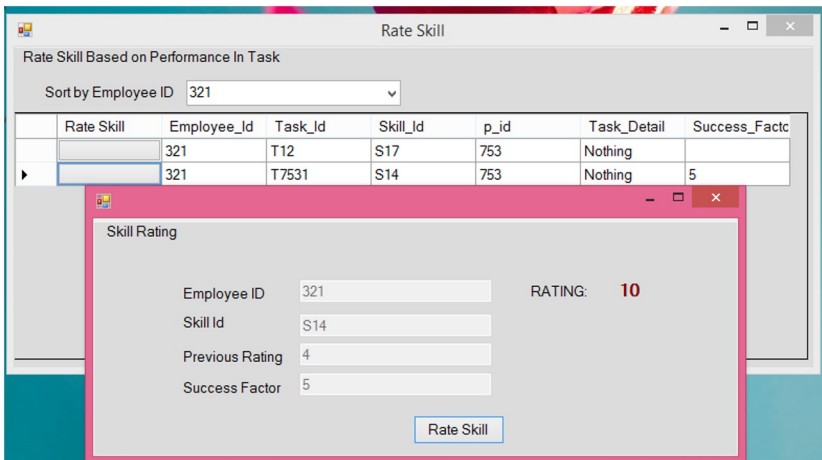

**Fig 6. Rate skills.**

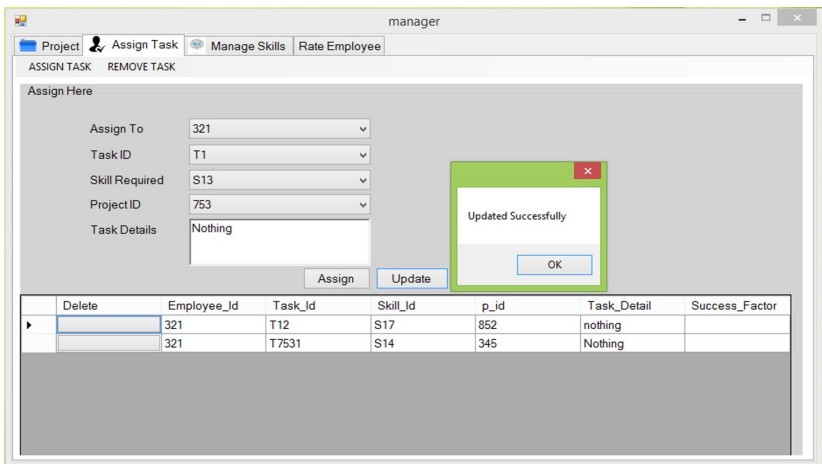

**Fig 7. Assign tasks.**

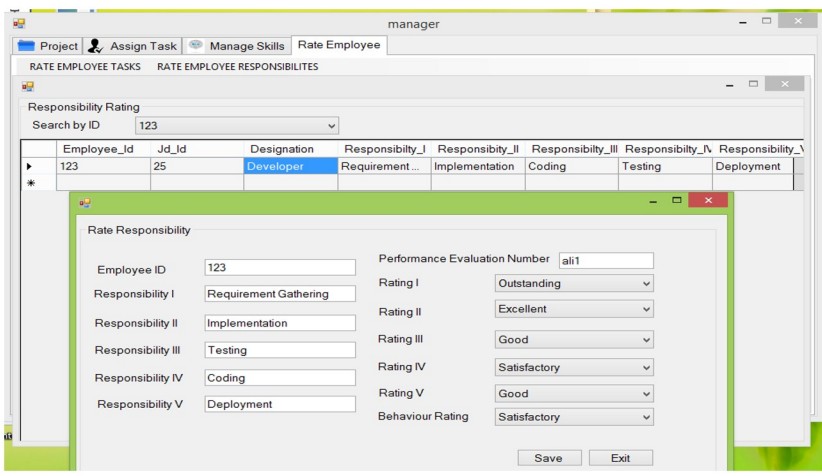

**Fig 8. Rate responsibilities.**

**5.2.2 Skill calculation.**   This module as given in Fig 5, is developed for the managers to calculate the skills of the employees. The skill calculation is based on the data entered in various forms that are filled by the user and manager. The data include educational record, training record, experience, project and task completion history, and performance of the employee. All information is used by the skill calculation module. This module increase or decreases the rating of employee skills automatically based on his/her performance on the current task(s).

**5.2.3 Skill updater.**   This module is used by the manager to update the particular skill of an employee based on his/her performance in the task. This updated information is used by the skill calculation module to rate the overall skills of employees.

**5.2.4 Manage project.**   This feature enables the manager to add, updates, view, and delete the project/project activities. The project manager will be able to view the work status of employees.

**5.2.5 Manage task.**   This feature allows the manager to add, update, view, and delete the tasks of a Project.

**5.2.6 Assign tasks.**   This feature allows the project manager to view skill profile before assigning tasks. This module facilitates PM to view how many tasks of employees are in progress or completed so that he can assign next tasks accordingly.

**5.2.7 Rate task.**   This feature allows the manager to rate an employee based on his performance on the task.

**5.2.8 Rate responsibilities.**   This feature allows the manager to rate employee responsibilities.

# 6 Tool evaluation / comparison

We have built the EBSM system based on the proposed algorithm as shown in Fig 4. The system evaluation process is performed using different methods which include: i) Questionnaire; ii) Facilitated Workshop and iii) Comparison of our prototype tool with existing tools. Detail of each part is given below:

## 6.1 Facilitated workshop

A facilitated workshop is arranged for the evaluation of the EBSM tool. 70 participants from the software development industry and academia are invited to attend the workshop. 38 participants attended the workshop. In the workshop, the authors give a live demonstration of the EBSM tool to participants. The demonstration includes a short presentation about the tool, installation of the tool, data entry, and usage demonstration. After the demonstration EBSM tool is given to the participants for evaluation. The participants have privileges to play the role of manager and employee at a time to assess the functionality of the tool.

At the end of the workshop, we distributed the evaluation questionnaire [46] to all participants and requested them to give their feedback. The questions are asked to get opinions and experience towards the EBSM tool in three major aspects including 1) the EBSM effectiveness in measuring and evaluating skills of the employees, 2) the EBSM tool applicability and need, and 3) the EBSM functionality. A total of 36 paper-based questionnaires are distributed to students, developers, project managers, and researchers. Collected data are classified into 3 groups: students, developers, and project managers. In the survey, respondents are asked to grade each parameter (usefulness, learning, usability, performance, effort, ease of installation, etc.) of the EBSM, according to their actual preferences in the working environment.

From the results of the survey [46], it is clear that our tool helps employees and organizations to easily manage and measure employee skills. The software is useful because it makes employees more effective in getting their job done [47]. No technical assistance is needed

because the system is user-friendly. Employee productivity and performance are increased as the system is simple to interact with. System functions are well integrated but do not have a built-in assistant facility which can be added in future versions.

In summary, respondents show a high level of satisfaction with the evaluation parameters of the tool. 58% respondents are very satisfied with the performance of the tool, 28% are satisfied and Only 3% are not satisfied with the performance. Overall 86% participants show a high level of satisfaction towards software performance. We have asked 5 sub-questions to measure the usefulness of the tool. 100% participants agreed that the EBSM tool is useful and effective for skills management. 50% respondents strongly agreed and 41% agreed about ease of use of this tool. In total 91% of the respondents agreed that the EBSM tool is easy to use. 89% of the participants show a high level of satisfaction. They want this tool at their workplace because this tool works as per their expectations. We have asked various questions during the survey to judge the usability of the system. The response of participants is healthy about the usability of the system and 72% of participants show their satisfaction as they can use this tool with very little effort. 94% of the respondents agreed that the EBSM system is useful at the job, 92% agreed that using this tool they can perform tasks quickly and 97% agreed that this system helps to increase their productivity. Effort expectancy of software is assessed and the respondents show a high level of satisfaction. 92% agreed that their interaction with the proposed system is clear, 97% agreed that the system is user-friendly and it is easy to learn the system. The attitude of survey respondents towards using this software is positive. 94% agreed using this system in an organization is a good idea. 92% agreed that using this system makes tasks easier to accomplish. We have asked questions to measure the Self-efficacy of the employee using the proposed system. 78% of the respondents agree that self-efficacy has been increased because an employee completes the task and needs no technical support. 97% agreed that the system installation is easy. 94% agreed that functionality is consistent with the interfaces. 100% of the respondents are satisfied with the overall performance of the system. We have uploaded the complete results of the survey on our GitHub project repository [46].

## 6.2 EBSM comparison with existing tools

We compare the features of the EBSM tool with three already developed tools named Skill Set Improvement [18], Skill Management [19] and Skill Manager Tool [20]. We have considered different research papers and after narrowing them down, we have identified papers that are closely related to our research. We have identified five features from research papers [18–20] for comparison. These features include algorithm, features, usage, development tool / language, and availability. Comparison results are given in Table 3. From the comparison, we can observe that the EBSM tool algorithm is better because it covers most of the attributes to calculate the skill. EBSM covers more features as compared with other software. Moreover, the usage of EBSM is simple and we have uploaded it on GitHub for future researchers.

## 7 Conclusion and future work

Assessment of human skills is vital; it determines the progress and profitability of businesses. Proper evaluation and assessment of human skills are very important because it is need for management. In order to evaluate human skills, organizations spend plenty of resources in terms of time, money, and efforts. We cannot neglect the importance of DSSs for skill calculation because of the rapid change in the software development industry. Our proposed system helps in decision-making and also fulfills the user expectations. The proposed system will overcome the problems stated in the literature. This system facilitates finding experts in an organization based on skillset. DSS using skill evaluation, facilitate managers as well as employees to

**Table 3. Comparison of proposed tool with existing tools.**

| Tools | Skill Set Improvement [18] | Skill Management [19] | Skills Manager Tool [20] | EBSM (Proposed) |
|---|---|---|---|---|
| Algorithm | Indexing technique | Resource Constraint scheduling problem | - | Skills Calculation |
| Features | Skill set, Role Played, No of years of Experience | Skill set, skills for tasks, number of tasks, cost of task, employee salary | Knowledge and level about subjects | Skill matrix, Experience Skill Set, Educational Skill Set, Rating, performance |
| Usage | Retrieves the resource with the best skill sets available in the database. The signature graph is constructed for the employee class. Signature matching is performed to find the closest match. | Genetic algorithm-based automated tools that assign the employee to project tasks optimally by considering various necessary configurations of duration and cost. | Employees can state the level of knowledge they have in different areas like Visual Basic, C#. Used for project staffing and it can help to find a qualified person for the task. | Calculate employee skills based on education, experience, project performance, and training. Update Skill matrix continuously by tracking and assessing the performance of individuals. |
| Dev. Tool | Java | - | Web based (HTML, PHP) | C#, Visual Studio |
| Availability | No | No | No | Available on GitHub |

cut down manual work. By using this system, managers will easily identify the employee's skills and then utilize these skills to increase productivity. This system can be utilized to find the knowledge gap in the organization as it successfully evaluates the skills and knowledge of the employees. Furthermore, System can be applied to a whole software development industry with little modifications. We would like our tool to be integrated with mainstream PM tools for the effective use of our approach in the future.

## Supporting information

**S1 Appendix. This file contains the data collection questioner which is used to collect data and requirements to develop EBSM software.** Results of the survey, Questioner used for the evaluation of developed tool. Finally, the results of evaluation survey are also enclosed in the file. (PDF)

## Acknowledgments

We thanks all the organizations and participants of the survey who helped us to give their valuable feedback to develop this framework.

## Author Contributions

**Conceptualization:** Nazia Bibi, Zeeshan Anwar.

**Data curation:** Nazia Bibi, Zeeshan Anwar.

**Formal analysis:** Nazia Bibi, Zeeshan Anwar.

**Investigation:** Tauseef Rana.

**Methodology:** Nazia Bibi, Zeeshan Anwar.

**Project administration:** Tauseef Rana.

**Software:** Nazia Bibi, Zeeshan Anwar.

**Supervision:** Tauseef Rana.

**Writing – original draft:** Nazia Bibi.

**Writing – review & editing:** Zeeshan Anwar, Tauseef Rana.

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
