## [Decision Letter · Decision Letter 0]

25 Nov 2020

PONE-D-20-22779

Expertise based Skills Management System to Support Resource Allocation

PLOS ONE

Dear Authors,

Thank you for submitting your manuscript to PLOS ONE. After careful consideration, we feel that it has merit but does not fully meet PLOS ONE’s publication criteria as it currently stands. Therefore, we invite you to submit a revised version of the manuscript that addresses the points raised during the review process.

Please see comments below.

We look forward to receiving your revised manuscript.

Kind regards,

Dejan Dragan, PhD

Academic Editor

PLOS ONE

Additional Editor Comments:

Two reviewers have conducted a review of this paper. They have detected a significant amount of deficiencies in the paper. The paper is very long and it does not emphasize the contribution. So, it is not clear how the work can contribute to which area. Accordingly, it is recommended that the paper is restructured to provide a strong focus on a certain aspects such as algorithm and the comparative evaluation been performed to show its effectiveness or accuracy. Moreover, the paper requires certain improvement especially on its main objective and refocusing to make it to show how the model can be useful to the targeted community. Based on these reviews, a major revision is required. AE DD

Journal Requirements:

2. Please upload a copy of the questionnaire used in the study as a supplemental file. Additionally, please note that PLOS ONE does not allow for the use of footnotes in its publications. As such, we ask you to remove all footnotes and move the information contained in them to the main text.

4. We note you have included a table to which you do not refer in the text of your manuscript. Please ensure that you refer to Table 2 in your text; if accepted, production will need this reference to link the reader to the Table.

Reviewers' comments:

Reviewer's Responses to Questions

**Comments to the Author**

1. Is the manuscript technically sound, and do the data support the conclusions?

Reviewer #1: Partly

Reviewer #2: Yes

2. Has the statistical analysis been performed appropriately and rigorously? 

Reviewer #1: N/A

Reviewer #2: N/A

3. Have the authors made all data underlying the findings in their manuscript fully available?

Reviewer #1: Yes

Reviewer #2: Yes

4. Is the manuscript presented in an intelligible fashion and written in standard English?

Reviewer #1: Yes

Reviewer #2: Yes

5. Review Comments to the Author

Reviewer #1: 1. The paper has presented one of the quite important model for the skill assessment. However, it lack critical review in that area.

2. The paper intends to propose a kind of a decision support tool that can help managers to perform skills assessment towards their employees. Hence, more literature on that area should be provided.

3. The basis and the literature review provided may not be sufficient to show how the model is developed.

4.The whole description of the proposed model and how the model is tested can be described in a technical report or a thesis. AT present , it is quite difficult to present the proposed model as a paper without having to say that there are still many missing information about the proposed model.

5. It is recommended that the paper just focused on one important aspect of the proposed model and the authors are able to give the details description of that particular aspect rather than having quite a long general description about the whole model development and evaluation.

Reviewer #2: The paper presents Skills Calculation Engine (SCE) framework for assigning the right worker with the right expertise to the righ task.

The authors claim that there is an extreme lack of available tools to support for monitoring and evaluating human skills, and conduct a survey across various software organizations to find out the best practices for skill management and to gather requirements from software development industry.

While this is a considerable effort, I find that this study did not conduct a thorough literature review as there is significant research on competence analysis and management such as:

Bohlouli, Mahdi, et al. "Competence assessment as an expert system for human resource management: A mathematical approach." Expert Systems with Applications 70 (2017): 83-102.

Could the authors elaborate on how the presented work is different from the work presented above where social competence is also considered?

And in any case, include competence management in the related work as they are related to the same topic.

Minor issues:

-I could not find any reference to Eq (1) in the manuscript (page 3).

-Short url is not working (page 3).

6. PLOS authors have the option to publish the peer review history of their article (what does this mean?). If published, this will include your full peer review and any attached files.

Reviewer #1: No

Reviewer #2: No

---

## [Author Response · Author response to Decision Letter 0]

27 Apr 2021

We are very thankful to you for performing an in-depth review of the manuscript and raising the important issues. The raised issues were very logical and their incorporation was essentially needed to transform a manuscript into a comprehensive study. These proposed changes really helped us to improve the quality of the paper. We have carefully understood the comments and have substantially revised our paper to address your concerns. Changes in the paper are highlighted as red color. The following table explains the incorporation status, response to each comment and reference to a part of manuscript where the suggestions have been incorporated. 

We again thank you and waiting for your response for publication of this paper in PLOS-One.

Sincerely,

---

## [Decision Letter · Decision Letter 1]

19 May 2021

PONE-D-20-22779R1

Expertise based Skills Management System to Support Resource Allocation

PLOS ONE

Dear Dr. Anwar,

Thank you for submitting your manuscript to PLOS ONE. After careful consideration, we feel that it has merit but does not fully meet PLOS ONE’s publication criteria as it currently stands. Therefore, we invite you to submit a revised version of the manuscript that addresses the points raised during the review process.

The Reviewers's comments have been addressed in the revised manuscript. However, the paper still needs a thorough professional mothertongue proofreading and an accurate check with the PLOS One formatting guidelines (for structure, style and reference format).

We look forward to receiving your revised manuscript.

Kind regards,

Alessandro Margherita

Academic Editor

PLOS ONE

Journal Requirements:

Reviewers' comments:

Reviewer's Responses to Questions

**Comments to the Author**

1. If the authors have adequately addressed your comments raised in a previous round of review and you feel that this manuscript is now acceptable for publication, you may indicate that here to bypass the “Comments to the Author” section, enter your conflict of interest statement in the “Confidential to Editor” section, and submit your "Accept" recommendation.

Reviewer #2: All comments have been addressed

2. Is the manuscript technically sound, and do the data support the conclusions?

Reviewer #2: Yes

3. Has the statistical analysis been performed appropriately and rigorously? 

Reviewer #2: N/A

4. Have the authors made all data underlying the findings in their manuscript fully available?

Reviewer #2: Yes

5. Is the manuscript presented in an intelligible fashion and written in standard English?

Reviewer #2: Yes

6. Review Comments to the Author

Reviewer #2: The article looks better and the authors addressed my concerns.

7. PLOS authors have the option to publish the peer review history of their article (what does this mean?). If published, this will include your full peer review and any attached files.

Reviewer #2: No

---

## [Author Response · Author response to Decision Letter 1]

9 Jul 2021

Reviewers have accepted all the changes that we made in previous major revision. In this minor revision reviewers have not proposed any change. We have corrected the language related issues and improved the format of paper.

---

## [Editor Report · Decision Letter 2]

28 Jul 2021

Expertise based Skills Management System to Support Resource Allocation

PONE-D-20-22779R2

Dear Dr. Anwar,

We’re pleased to inform you that your manuscript has been judged scientifically suitable for publication and will be formally accepted for publication once it meets all outstanding technical requirements.

Kind regards,

Alessandro Margherita

Academic Editor

PLOS ONE
---

## [Editor Report · Acceptance letter]

6 Aug 2021

PONE-D-20-22779R2 

Expertise based Skills Management System to Support Resource Allocation 

Dear Dr. Anwar:

I'm pleased to inform you that your manuscript has been deemed suitable for publication in PLOS ONE. Congratulations! Your manuscript is now with our production department. 

Kind regards, 

on behalf of

Dr. Alessandro Margherita 

Academic Editor

PLOS ONE